# Nanoporous–Crystalline Poly(2,6-dimethyl-1,4-phenylene)oxide Aerogels with Selectively Sulfonated Amorphous Phase for Fast VOC Sorption from Water

**DOI:** 10.3390/ma15051947

**Published:** 2022-03-05

**Authors:** Marina Pellegrino, Adriano Fiumarella, Alma Moretta, Christophe Daniel, Marco Trifuoggi, Anna Borriello, Vincenzo Venditto

**Affiliations:** 1Dipartimento di Chimica e Biologia A. Zambelli, Unità INSTM, Università Degli Studi di Salerno, Via Giovanni Paolo II, 84084 Fisciano, Salerno, Italy; mpellegrino@unisa.it (M.P.); a.fiumarella@studenti.unisa.it (A.F.); almamoretta@gmail.com (A.M.); vvenditto@unisa.it (V.V.); 2Dipartimento di Scienze Chimiche, Università Degli Studi di Napoli Federico II, Complesso Universitario di Monte Sant’Angelo, Via Cintia, 21, 80126 Napoli, Italy; marco.trifuoggi@unina.it; 3Institute for Polymers, Composites and Biomaterials, National Research Council of Italy, P.le Fermi, 1, 80055 Portici, Naples, Italy

**Keywords:** poly(2,6-dimethyl-1,4-phenylene)oxide, aerogel, solid-state sulfonation

## Abstract

This paper describes the preparation and characterization of poly(2,6-dimethyl-1,4-phenylene)oxide (PPO) highly porous monolithic aerogels with a hydrophobic nanoporous–crystalline phase and a hydrophilic sulfonated amorphous phase. The sulfonated aerogels were obtained by the sulfonation of PPO physical gels, followed by the supercritical CO_2_ extraction of solvents. WAXD and FTIR analysis showed that the nanoporous–crystalline phase was preserved for a degree of sulfonation up to c.a. 35%, allowing a highly volatile organic compound (VOC) sorption capacity. The sulfonated PPO aerogels exhibited a high water sorption capacity, with a water uptake of up to 500 wt%, and faster VOC sorption kinetics from water with respect to unsulfonated aerogels.

## 1. Introduction

Aerogels, usually produced by wet gel drying, represent a unique class of materials. Many examples of aerogels have been reported in the literature, based on organic or inorganic material, such as clays and silica [1,2,3,4,5,6] or polymers [7,8,9,10,11,12,13,14,15,16,17,18,19].

According to the literature, monolithic polymeric aerogels are most commonly obtained with chemically crosslinked polymers (e.g., resorcinol–formaldehyde and melamine–formaldehyde [7,8,9], polyurethane [10], polyimide and polyamide [11,12], and modified cellulose [13]). However, during the last decade, the preparation of robust monolithic polymeric aerogels with a three-dimensional network based on crystalline regions rather than covalent bonds has also been reported (for many polymers, such as syndiotactic polystyrene [14], poly(L-lactic acid) [15], poly(vinylidene fluoride) and poly(vinylidene fluoride-co-hexafluoropropylene) [16,17], poly(4-methyl-1-pentene) [18], and poly(2,6-dimethyl-1,4-phenylene)oxide [19]).

Among these different polymers, particularly interesting are poly(2,6-dimethyl-1,4-phenylene)oxide and syndiotactic polystyrene (respectively, PPO and s-PS), which are the only polymers that form aerogels characterized by nanoporous–crystalline phases (or ultramicroporous, according to IUPAC classification) alongside the typical meso- and macropores common to all aerogels [14,19]. As a result of the nanoporous–crystalline phases, PPO and s-PS aerogels exhibit particularly high BET surface areas (up to 750 m^2^/g for PPO). Furthermore, these aerogels show a high capacity for absorbing traces of volatile organic compounds (VOCs) in water or air [14,19], which make them interesting from the perspective of air and water purification applications [20,21].

Sulfonated s-PS monolithic aerogels with a hydrophobic nanoporous–crystalline phase and a hydrophilic sulfonated amorphous phase have been prepared through the sulfonation of s-PS physical gels, followed by the supercritical CO_2_ extraction of the solvent [22]. As a result of the high and fast water uptake, these sulfonated s-PS monolithic aerogels present VOC sorption kinetics of up to three orders of magnitude higher than unsulfonated aerogels and five orders of magnitude higher than sulfonated s-PS film [23,24]. However, it has also been observed that, due to sulfonation, the aerogel uptake capacity significantly decreases.

With respect to s-PS aerogels, PPO aerogels present a higher uptake capacity towards many organic compounds such as benzene; carbon tetrachloride; and several alkanes such as n-pentane, heptane, and decane [19]. The higher melting temperature of PPO nanoporous–crystalline phases [25,26] also ensures a higher thermal stability compared to s-PS aerogels, which in turn allows an easy and cheap regeneration by thermal treatments [19].

In analogy with s-PS, sulfonated PPO monolithic aerogels were also obtained in order to assess the possible improvement of the VOC sorption properties. In this paper, the preparation and characterization of sulfonated PPO aerogels with nanoporous–crystalline phases are reported for the first time. It is shown that PPO aerogels with a porosity higher than 85% and a sulfonation degree of 35% can be obtained by the sulfonation of PPO gels, followed by solvent extraction with supercritical CO_2_.

X-ray diffraction analysis (WAXD), infrared spectroscopy (FTIR), and scanning electron microscopy (SEM) were applied for the characterization of the obtained PPO aerogels with a hydrophobic nanoporous–crystalline phase and a hydrophilic sulfonated amorphous phase. Moreover, nitrogen sorption–desorption measurements and investigations into tetrachloroethylene (PCE) sorption from vapor phase and water were carried out using PPO monolithic aerogels with different degrees of sulfonation.

The sorption tests showed that sulfonated PPO aerogels are characterized by a high PCE uptake associated with faster sorption kinetics than unsulfonated PPO aerogels. These transport properties make sulfonated PPO aerogels particularly appealing for potential use as VOC sorbents for water purification.

## 2. Materials and Methods

### 2.1. Materials

For this study, PPO (Ultra High P6130 grade, Mw = 350,000 g mol^−1^) provided by Sabic (Sittard, The Netherlands) was used. All solvents and reagents were supplied by Aldrich (Milan, Italy) and used with no further purification.

### 2.2. Sample Preparation Procedure

The PPO gels were produced by dissolution at c.a. 130 °C in hermetically sealed test tubes of PPO powder in methyl benzoate (solvent:polymer 9:1 wt ratio) followed by gelation at room temperature.

A mild sulfonation reagent (sulfo dodecanoate) was prepared according to previously reported procedures [23,27] by mixing an excess of dodecanoic acid (lauric acid, ≥98%, 4.8 × 10^−2^ mol) with chlorosulfonic acid (ClSO_3_H, 99%, 3.0 × 10^−2^ mol) at room temperature for 24 h under nitrogen atmosphere.
CH_3_(CH_2_)_10_COOH + ClSO_3_H → CH_3_(CH_2_)_10_COOSO_3_H + HCl

PPO gel cylinders (diameter c.a. 6 mm, length c.a. 30 mm, weight c.a. 1.0 g) were dipped for c.a. 24 h into solutions prepared with a certain amount of sulfonating reagent and 50 mL of cyclohexane, in order to obtain the gel swelled with the sulfonating reagent. After removing the swelled PPO gel cylinders from the sulfonating solution and soaking them in cyclohexane, thermal activation of the sulfonation reaction was conducted at T = 50 °C (Figure 1).

By using different heating times and different amounts of sulfonating reagent in the solution, it was possible to vary the sulfonation degree of the obtained samples (i.e., aerogel with 22% sulfonation degree was obtained by using 4 g of sulfonating reagent and 2 h heating).

The sulfonation reagents were removed by washing the sulfonated gels at 25 °C in c.a. 100 mL of ethanol under mild stirring for 3 h. Finally, solvent extraction with scCO_2_ was the last step to obtain the final sulfonated aerogels. The absence of sulfonation reagent residuals in the aerogels was ensured by FTIR.

The benefit of scCO_2_ extraction is the absence of surface tension. During the solvent removal process, a supercritical solution is formed between the solvent and the supercritical CO_2_, and it is possible to extract the solvent without collapsing the structure [28].

An SFX 200 supercritical CO_2_ extractor (ISCO Inc., Lincoln, NE, USA) was used for the solvent extraction (P = 200 bar; T = 40 °C; extraction time = 180 min).

In the case of monolithic aerogels characterized by a regular shape, it is possible to estimate the aerogel total porosity (macroporosity, mesoporosity, microporosity) from the mass:volume ratio. In this case, the aerogel percentage of porosity, P, can be determined as:(1)$$P=100\left(1-\frac{\rho_{app}}{\rho_{pol}}\right)$$
where *ρ_app_* is the aerogel apparent density, obtained from the aerogel mass:volume ratio, and *ρ_pol_* is the polymer matrix density (equal to c.a. 0.99 g/cm^3^ for a semicrystalline nanoporous α-form PPO with a crystallinity of 50% and considering a density of amorphous PPO of 1.04–1.06 g/cm^3^ and a density of α-form PPO equal to 0.93 g/cm^3^ [29]).

### 2.3. Techniques

A Flash EA 1112 analyzer (Thermo Fisher Scientific, Waltham, MA, USA) was used to perform elemental analysis of aerogel sections, in order to determine the aerogel sulfonation degree. The determined sulfonation degree (S) is expressed as a molar percent of sulfonated monomeric units:S = 100 × ((sulfur moles by elemental analysis)/(total moles of polymer units))(2)

A Stereoscan S440 scanning electron microscope (Leica Cambridge, Milton Keynes, UK) equipped with a probe for energy-dispersive scanning (EDS), was used to obtain electron micrographs for evaluation of the sulfonation degree along the entire thickness of the sample. In order to reach the internal parts of the aerogels for the SEM–EDS experiments, a cryogenic fracture of the samples was carried out using liquid N_2_. All the specimens were subjected, before imaging, to gold coating (through a VCR high-resolution indirect ion-beam-sputtering system).

A D8 Advance automatic diffractometer (Bruker, Karlsruhe, Germany) was utilized to determine wide-angle X-ray diffraction (WAXD) patterns, in reflection, using Ni-filtered CuKα radiation. By resolving the WAXD pattern into two areas, A_cr_ and A_am,_ respectively proportional to the crystalline and amorphous weight fractions, it was possible to determine a crystallinity index (X_cr_), applying the following expression:X_cr_ = [A_cr_/(A_cr_ + A_am_)] × 100(3)

A Tensor 27 spectrometer (Bruker, Karlsruhe, Germany), equipped with a Ge/KBr beam splitter and a deuterated triglycine sulphate (DTGS) detector, was used to obtain infrared spectra at a resolution of 2.0 cm^−1^ (a He–Ne laser was used to calibrate internally the frequency scale to 0.01 cm^−1^; noise reduction was achieved by averaging 32 scan signals).

A Nova 4200e instrument (Quantachrome, Odelzhausen, Germany) was used for surface area measurement of both sulfonated and unsulfonated aerogels by N_2_ adsorption at 77 K. Aerogel samples were subjected to degassing under vacuum (for 24 h at 45 °C) before the adsorption measurement. The Brunauer–Emmet–Teller (BET) method was used for calculation of the surface area values.

A VTI-SA symmetrical vapor sorption analyzer (TA Instruments, New Castle, DE, USA) was utilized to measure the vapor sorption at 35 °C. The fraction of dry gas and the fraction of gas going through the organic solvent evaporator were used to determine the organic vapor concentration in the gas stream reaching the sample.

Tetrachloroethylene (PCE) sorption from aqueous solutions was investigated by measurements of FTIR absorbance of PCE and PPO peaks using calibration curves.

## 3. Results and Discussion

### 3.1. Sulfonated PPO Aerogel Preparation

The dimensions and shape of the starting PPO gels did not substantially change during the successive sulfonation and solvent extraction steps, allowing the production of aerogel monoliths as shown in Figure 1.

Figure 2 reports typical SEM images of an unsulfonated PPO aerogel and of sulfonated PPO aerogels with S = 9% and S = 25%, obtained from gels prepared with methyl benzoate at C_pol_ = 0.1 g/g.

In Figure 2, we can observe that the PPO aerogel morphology was not modified by the gel-state sulfonation, and the fibrillar morphology obtained in the unsulfonated aerogels was also maintained in the sulfonated aerogels.

### 3.2. Structural Characterization of Sulfonated PPO Aerogels

The aerogel sulfonation degree can be tuned by changing the amount of sulfonating reagent, the gel swelling time, or the heating reaction time.

As an example, Figure 3A reports the FTIR spectra in the wavenumber range of 800–400 cm^−1^ for unsulfonated and sulfonated PPO aerogels obtained for different swelling times and heating reaction times.

For the sulfonated PPO aerogel, we can observe a peak at c.a. 670 cm^−1^ that has been assigned to the C–S stretching vibration [30]. The absorbance of this peak increases with an increasing swelling time and heating reaction time, thus indicating an increase in the sulfonation degree.

It is also worth observing that the IR bands at 773 and 414 cm^−1^, characteristic of the nanoporous–crystalline α-form [29], appear to remain substantially constant after sulfonation.

Figure 3B reports the variation of the normalized absorbance ratio of the 773 and 414 cm^−1^ bands as a function of the degree of sulfonation.

The results clearly show that the ratio remained constant within the experimental uncertainties for a degree of sulfonation up to c.a. 35%. This behavior indicates that the crystallinity of the gel is fully preserved during sulfonation. Thus, the sulfonation of PPO gels occurs mainly in the amorphous phase (also called blocky sulfonation [31]).

A structural characterization of the sulfonated aerogels was carried out, and the WAXD patterns of the PPO aerogels with a sulfonation degree up to c.a. S = 31% are shown in Figure 4.

It can be observed that both the unsulfonated and sulfonated aerogels show strong reflections at 2θ = 4.5, 7.2, 11.3, 15, and 18.3°, which is typical of the PPO nanoporous–crystalline α-form [29].

The WAXD diffraction patterns also indicate that the degree of crystallinity was substantially unchanged in the investigated range of sulfonation and that a degree of crystallinity of c.a. 50% was obtained for the unsulfonated and sulfonated aerogels, therefore confirming that only the aerogel amorphous phase was sulfonated without any loss of crystallinity.

In addition, it is worth observing that the conservation of the gel crystalline junctions allowed the preservation of the gel dimensions during sulfonation, and only a small decrease in the apparent porosity was observed (column four of Table 1).

### 3.3. Surface Area Characterization of Sulfonated Aerogels

The surface areas (BET) of the unsulfonated and sulfonated aerogels determined from nitrogen sorption measurements at 77K, which are reported in column five of Table 1, are compared in Figure 5.

A progressive decrease in the BET surface area with the degree of sulfonation (S) can be clearly observed. In detail, the BET surface area of an aerogel with S = 35.4% was 296 m^2^/g, while for the unsulfonated aerogel the BTE surface area was 518 m^2^/g.

Both the FTIR and WAXD analyses, reported in previous sections, demonstrated that the crystallinity degree of sulfonated aerogels remained almost unchanged. As a consequence, the decrease in the BET surface area cannot be due to a decrease in the amount of crystalline nanopores but may be due to a decrease in the amount of micropores located between the polymer fibrils or within the polymer amorphous phase.

### 3.4. Sorption Properties of Sulfonated PPO Aerogels

#### 3.4.1. Water Sorption

Figure 6 reports the liquid water uptake of the sulfonated PPO aerogels with different degrees of sulfonation. We can observe, as expected, an increase in water uptake in the sulfonated PPO aerogels with an increasing sulfonation degree. A water uptake of c.a. 500 wt% was obtained for the PPO aerogel with S = 36%, while the unsulfonated PPO aerogel was totally hydrophobic.

#### 3.4.2. VOC Sorption Properties

For the purpose of assessing the effect of the sulfonation degree on the sorption capacity of PPO aerogels, the uptake of tetrachloroethylene (PCE) (a widespread pollutant of groundwater) was investigated for low PCE-vapor pressures.

The sorption uptake of PCE vapor at 35 °C in the unsulfonated and sulfonated PPO aerogels with different degrees of sulfonation are compared in Figure 7. For the sake of discussion, the PCE sorption properties of the unsulfonated and sulfonated s-PS aerogels are also shown.

Figure 7 shows that the PCE-vapor sorption capacity remained substantially constant for the unsulfonated and sulfonated PPO aerogels with a sulfonation degree up to 31%.

For s-PS, we can note that the aerogel sorption capacity of the unsufonated aerogel was comparable to that of the unsulfonated PPO aerogel, but the PCE uptake progressively decreased as the degree of sulfonation increased. In particular, at P/P_0_ = 0.01, the PCE uptake in the unsulfonated s-PS aerogel was c.a. 12.3 wt%, while for the s-PS aerogels with degrees of sulfonation of S = 8% and S = 32%, the PCE uptake was 6.4 wt% and 2 wt%, respectively.

These results can be explained by a continuous decrease in the degree of crystallinity with an increase in the sulfonation degree for s-PS aerogels [24], while for PPO aerogels, the degree of crystallinity remained substantially constant for the range of sulfonation degree investigated; thus, the uptake capacity of PPO aerogels did not vary.

A preliminary study of VOC sorption from water was also carried out. The PCE sorption kinetics from a 10 ppm aqueous solution for an unsulfonated PPO aerogel and a sulfonated (S = 9%) PPO aerogel are compared in Figure 8.

The results reported in Figure 8 show that the sulfonated PPO aerogel presented a higher uptake capacity and faster sorption kinetics with respect to the unsulfonated aerogel. These results are possibly due to the water sorption in the hydrophilic sulfonated amorphous phase of the PPO aerogel, which allows a faster diffusivity of PCE within the polymer fibers and a higher uptake within the PPO nanoporous–crystalline phase.

## 4. Conclusions

Sulfonated semicrystalline PPO monolithic aerogels, with an apparent porosity of about 85%, were obtained for the first time through a procedure of the sulfonation of gels in mild conditions and subsequent solvent extraction with supercritical carbon dioxide. This procedure led to aerogels with a hydrophobic nanoporous–crystalline phase and a hydrophilic sulfonated amorphous phase, without any loss of crystallinity for sulfonation up to c.a. 35%.

Water sorption measurements in sulfonated aerogels showed a high water uptake, increasing with the sulfonation degree. Water uptakes greater than 500 wt% were observed for aerogels with a sulfonation degree of c.a. 35%.

PCE sorption tests from vapor phase showed that, conversely to s-PS aerogels, the uptake capacity of sulfonated PPO aerogels remained constant even for a high degree of sulfonation. This result was attributed to the sulfonation of the aerogel amorphous phase and the maintaining of the nanoporous–crystalline phase. The high uptake capacity, associated with fast sorption kinetics, makes sulfonated PPO aerogels particularly appealing for potential applications in water purification devices.

## Data Availability

Not applicable.

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
