# Peer review of "Nanoporous–Crystalline Poly(2,6-dimethyl-1,4-phenylene)oxide Aerogels with Selectively Sulfonated Amorphous Phase for Fast VOC Sorption from Water"

_materials, 2022, doi:10.3390/ma15051947_

Round 1

Reviewer 1 Report

The present work deals with the functionalization of porous monolithic aerogels prepared from PPO. Authors argue that the sulfonation process occurs mostly in the amorphous region of the material, and therefore, the obtained samples exhibit an amphiphilic nature since crystalline regions remain hydrophobic, while amorphous (due to sulfonate groups) acquire a hydrophilic property. Obtained materials were tested in terms of water and volatile organic compounds uptake, showing good results. However, this manuscript can be notably improved in both, quality of presentation as well as scientific soundness. I would recommend this paper for publication after minor revisions.

Here I highlight some points to be considered:

Abstract section:

Line 19: I don't like the word "selective", it is true, some results support this, but the method itself is not selective, I just happen mainly in the amorphous fraction of the sample, and of course,I would not discard surface sulfonation in crystalline regions at some extent

Line 23: VOC is not defined previously in the text

Introduction:

line 30: references should be added regarding those polymers reported in the literature

line 57: I recommend changing "for" --> towards

line 63: "in this paper THE preparation and characterization of sulfonated PPO aerogels with nanoporous-crystalline phases is reported."

line 69: In the abstract authors mention hydrophobic and hydrophilic. I recommend following the same idea along the text

Materials and Methods

line 80-84: Please use straightforward descriptions. Also, fix the redaction of the text. Indicate the exact temperature at which was boiled the solvent 

line 85-95: i would really appreciate seeing some scheme about the chemistry involved in the process (reactions, etc).

What is dodecanoic acid used for?

In addition, I´m concerned regarding the cleaning process of your sulfonated gels, how you can ensure the chemical reaction, and not simple occlusion of sulfonate compounds. I notice that in line 96 you mention the cleaning process was using ethanol. Therefore, I would like to see a more detailed explanation of the process. Temperature, time, shaking or stirring, amount of solvent etc. This is a crucial step in your process. 

line 109: authors use a different symbol for the apparent density

line 110: how do you calculate the polymer matrix density?

Include in this section a more detailed explanation of the supercritical drying process.

Regarding equation 2: Any reference for this equation, it makes sense, however, how do you calculate the moles of monomeric units when you don't know how many units are already modified?

Results and discussion:

Line 145-147: This was already mentioned before

Figure 1: Figure can be notably improved. Pictures in vertical would help for comparison purposes. Also, the edges of monoliths are not in the zero position. Any idea for color changes observed?

Figure 2: 2A) resolution should be improved. Scale bars should be equal

Line 168-169: Were these parameters studied in this work? if not, add some references to this statement

Line 171 - 172: Fix the redaction of "different swelling times heating reaction times are reported in Figure 3A"

Figure 3: Please, increase substantially the quality of images, also, the caption of this figure is too long

line 195: replace "is reported" by are shown

Figure 4: increase the quality of the image. Also, numbers are not clear and different styles have been used during the preparation of this figure. For example, 25 % seems to be in bold format

Table 1: What was the idea of not stating a clear order in the Table? Ordering samples from low to high (or high to low) would facilitate the understanding of the reading audience.

line 215: it would be useful to explain where these values (1.49 and 0.99) come from

Figure 5: increase the quality. Regarding "This decrease of the BET surface area values may be due to a decrease of the amount of micropores located within the polymer fibrils or within the polymer amorphous phase." Any idea why this?

do you have any idea about the crystalline/amorphous proportion in samples?

Figure 6: What was the temperature of the experiment?

Increase the quality of figures 7 and 8

line 268: fix the format of reference 24 

conclusions: 

Line 295: fix "nanoporous"

Reviewer 2 Report

In the manuscript entitled “Nanoporous-Crystalline Poly(2,6-dimethyl-1,4-phenylene)oxide aerogels with selectively sulfonated amorphous phase for fast VOCs sorption from water” authors successfully prepared poly(2,6-dimethyl-1,4-phenylene)oxide highly porous monolithic aerogels with a hydrophobic nanoporous-crystalline phase and a hydrophilic sulfonated amorphous phase, possessing a high VOC sorption capacity.

X-ray diffraction analysis, FTIR, and SEM have been applied for the characterization of the obtained hydrophilic sulfonated PPO aerogels. The degree of sulfonation did not affect the morphology and crystallinity of PPO aerogel. Nitrogen sorption-desorption measurements were conducted, resulting in decreased BET surfaces in sulfonated samples. The study on VOCs sorption from vapor phase and water was carried out using PPO monolithic aerogels with different degrees of sulfonation.

The results might be interesting for the publications after minor revisions:

  1. The highlight of the paper must be clearly stated. Although one can directly see from the title that such materials can serve as VOC sorbents, the novelty and significance of the results are not clearly emphasized.
  2. The manuscript is very hard to read. The sentences are too long. English presentation is very poor. Authors should completely revise the English presentation of the paper.
  3. Figures are of a very poor resolution. It should be corrected.
  4. Legends should be included in Figure 7, not in the caption.
